



# Ozone-induced gross primary productivity reductions over European forests inferred from satellite observations

Jasdeep Singh Anand[1,2], Alessandro Anav[3], Marcello Vitale[4], Daniele Peano[5], Nadine Unger[6], Xu Yue[7], Robert J. Parker[1,2], and Hartmut Boesch[1,2]

[1]Earth Observation Science, School of Physics and Astronomy, University of Leicester, Leicester, UK
[2]National Centre for Earth Observation (NCEO), University of Leicester, Leicester, UK
[3]Energy and Sustainable Economic Development, National Agency for New Technologies (ENEA), Rome, Italy
[4]Department of Environmental Biology, Sapienza University of Rome, Rome, Italy
[5]Fondazione Centro Euro-Mediterraneo sui Cambiamenti Climatici, CSP, Bologna, Italy
[6]College of Engineering, Mathematics and Physical Sciences, University of Exeter, UK
[7]Jiangsu Key Laboratory of Atmospheric Environment Monitoring and Pollution Control, Collaborative Innovation Center of Atmospheric Environment and Equipment Technology, School of Environmental Science and Engineering, Nanjing University of Information Science Technology (NUIST), Nanjing, 210044, China

**Correspondence:** Jasdeep Singh Anand, jsa13@le.ac.uk

**Abstract.**

Tropospheric $O_3$ damages leaves and directly inhibits photosynthesis, posing a threat to terrestrial carbon sinks. Previous investigations have mostly relied on sparse in-situ data or simulations using land surface models. This work is the first to use satellite data to quantify the effect of $O_3$ exposure on gross primary productivity (GPP). $O_3$-induced GPP reductions were estimated to vary between 0.36-9.55% across European forests along a North-South transect between 2003-2015, in line with prior estimates. No significant temporal trend could be determined over most of Europe, while Random Forest analysis (RFA) shows that soil moisture is a significant variable governing GPP reductions over the Mediterranean. Comparisons between this work and GPP reductions simulated by the Yale Interactive Biosphere (YIBs) model suggest that satellite-based estimates over the Mediterranean region may be biased by +12%, potentially because of differences in modelling stomatal sensitivity to soil moisture and prior $O_3$ exposure. This work has demonstrated for the first time that satellite-based datasets can be leveraged to assess the impact of $O_3$ on the terrestrial carbon sink, which are comparable with in-situ or model-based analyses.

## 1 Introduction

Terrestrial ecosystems serve an important role in regulating atmospheric carbon dioxide ($CO_2$) concentrations, because they absorb and sequester $CO_2$ via photosynthesis (Gross Primary Productivity, GPP). Because of this, vegetation uptake of $CO_2$ is regarded as a major sink of anthropogenic carbon emissions, having removed about 30% of cumulative global emissions





between 1850-2018 (Friedlingstein et al., 2019). In particular, the size of the carbon uptake by vegetation in Europe has been estimated to be 3.9-5.8 PgC yr$^{-1}$ by Beer et al. (2007), while Wißkirchen et al. (2013) estimated it to be 2.5 PgC yr$^{-1}$.

The conservation and enhancement of terrestrial ecosystems over the next century forms an essential component in mitigation strategies to avoid dangerous climate change (Rogelj et al., 2018); two-thirds of signatories to the Paris Climate Agreement have indicated that they will use the terrestrial sink to meet their cumulative emission reduction targets (Grassi et al., 2017). However, existing models of mitigation pathways do not take into account indirect feedbacks in the carbon sink caused by pollution-induced vegetation damage (Rogelj et al., 2018).

A significant factor affecting terrestrial ecosystem GPP in recent decades is the effect of tropospheric ozone (O$_3$) on vegetation. In the troposphere O$_3$ is produced through complex chemical reactions involving anthropogenic emissions of precursor species (Myhre et al., 2013), such as nitrogen oxides (NO$_x$), volatile organic compounds (VOCs), and methane (CH$_4$).

As a powerful oxidant, O$_3$ damages leaf cells and inhibits photosynthesis when absorbed by plants through the stomata, accelerating leaf senescence (Wittig et al., 2009; Ainsworth et al., 2012). Additionally, stomata damaged by O$_3$ exposure are less capable of opening and closing in response to heat stress, causing dehydration and further injury (i.e. stomatal sluggishness, see: Wilkinson and Davies, 2009; Hoshika et al., 2015). As a result, prolonged O$_3$ exposure will cause a decline in GPP and so reduce the total carbon sequestered.

The magnitude of damage caused by O$_3$ exposure to forests is dependent on genotype and micro-climate (Matyssek et al., 2010), which complicates efforts to quantify O$_3$-induced GPP reductions and their consequent impact on the terrestrial carbon sink.

Prior investigations have either analysed long-term observations of carbon and O$_3$ fluxes from eddy covariance towers (e.g. Fares et al., 2013; Yue et al., 2016; Verryckt et al., 2017), or simulations run using land surface models (e.g. Sitch et al., 2007; Yue and Unger, 2014; Oliver et al., 2018). While most studies agree that O$_3$ exposure results in significant reductions in GPP, the estimated magnitude varies with measurement location or assumptions used in the models. For instance, Oliver et al. (2018) concluded that O$_3$-induced GPP reductions over Europe ranged between 2%-8% for boreal regions and between 10%-20% for temperate regions. However, Verryckt et al. (2017) found that no statistically significant effect could be determined for a single Belgian pine forest.

The effect of O$_3$ on the terrestrial carbon sink over the next century and the resulting effect on climate mitigation strategies has been the focus of much speculation. While emission control policies have been successful in reducing O$_3$ precursor concentrations in Europe over the last decade (EEA, 2020), this does not necessarily result in lower O$_3$ concentrations, as O$_3$ formation also depends on chemical regimes (Beekmann and Vautard, 2010). Thus, while O$_3$ concentrations since the year 2000 have decreased to an extent following these emission reductions (Proietti et al., 2020), it is unlikely that these moderate reductions in anthropogenic precursor emissions will adequately decrease future vegetation exposure to O$_3$ (Fuhrer et al., 2016; Sicard et al., 2017).

The effect of O$_3$ exposure on vegetation is species- and context-specific, and significant uncertainties remain. Long-term monitoring of vegetation responses to different O$_3$ concentrations and climate conditions are necessary to better constrain these estimates (Paoletti et al., 2019; Ainsworth et al., 2020).





Both measurement and simulation studies currently rely on limited datasets of vegetation response to $O_3$ exposure. In Europe, forest $O_3$ exposure and its effect on defoliation and leaf injury are measured at several permanent monitoring sites as part of the MOnitoring ozone injury for seTTing new critical LEvelS (MOTTLES; Paoletti et al., 2019) network. However, these sites

exist only in central and Mediterranean Europe, and have only been in operation since 2016. Similarly, eddy covariance towers, such as those that make up the global FLUXNET dataset (Pastorello et al., 2020) can also be employed to investigate the effect of $O_3$ exposure on GPP, but these also have sparse spatial sampling. While land surface models can be used to provide more expansive estimates of $O_3$-induced GPP reductions, these are calibrated using data from fumigation or field studies which may not be fully representative of regional ambient conditions.

In recent years satellite observations of atmospheric and terrestrial properties have become increasingly useful to our understanding of the global climate system, offering reliable coverage of remote regions not typically covered by in-situ stations. Because of the daily overpass time and high spatial resolution of many satellite instruments, reliable long-term satellite datasets now exist for essential climate variables (ECVs) relative to this field, such as tropospheric $O_3$ (e.g. Ziemke et al., 2019), land cover (e.g. Bontemps et al., 2013), and soil hydrology (e.g. Dorigo et al., 2017).

Such datasets have previously been assimilated into land surface models to improve their predictions (e.g. Orth et al., 2017). Some prior investigations into $O_3$-induced vegetation damage have also made use of satellite data. For instance, Fishman et al. (2010) used satellite-derived tropospheric $O_3$ column data as part of a regression model to estimate the annual soybean crop yield lost over the Midwest of the USA caused by $O_3$ exposure. Similarly, Proietti et al. (2016) combined GPP data from the Moderate resolution Imaging Spectroradiometer (MODIS; Zhao et al., 2005) with in-situ $O_3$ measurements in order to

estimate the $O_3$-induced GPP reduction over specific European sites.

However, these investigations have still relied on sparse in-situ measurements to provide necessary information, such as $O_3$ concentration and soil moisture. This work aims to provide the first estimates of European $O_3$-induced GPP reductions where satellite-based datasets are used to inform near-surface $O_3$ concentration and meteorology governing GPP. Such estimates would be useful to provide independent verification to land surface models that previously would have been limited by large-

scale measurement coverage.

## 2 Datasets and Methods

### 2.1 Study area and time period

For this work, $O_3$-induced GPP reduction estimates are calculated for European forests (latitude: 36°-60°N, longitude: 10°W-35°E). Only forested regions (deciduous or coniferous) were analysed in order to simplify calculations. GPP reduction es-

timates were calculated for 2003-2015, which was the overlapping time period available for all satellite datasets discussed herein. Additionally, only warmer months (April-September) were analysed for each year, as this time period is thought to encompass all possible growing season durations across Europe (Mills et al., 2017).





## 2.2 Datasets

Excluding soil moisture and meteorology, the datasets used in this work were regridded from their native resolutions to the
spatial grid used by the ERA5 reanalysis dataset (see Section 2.2.2), using a conservative regridding algorithm provided by the
xESMF software library developed by Zhuang et al. (2020).

### 2.2.1 $O_3$

Satellite observations of tropospheric $O_3$ are often either vertical profiles retrieved from nadir observations of ultraviolet (UV)
radiation (e.g. Miles et al., 2015), or integrated tropospheric columns derived by first estimating and then removing the
stratospheric component. The stratospheric component can be derived either through differencing total $O_3$ columns retrieved
in cloudy and cloud-free pixels (Heue et al., 2016), partitioning of the $O_3$ profile after estimating the tropopause height (e.g.
Miles et al., 2015), or from co-located measurements by limb-observing instruments (e.g. Ziemke et al., 2006).

However, inferring near-surface $O_3$ concentrations from these satellite datasets for this work is nontrivial. $O_3$ profiles re-
trieved from UV measurements have peak sensitivity in the lower troposphere (i.e. below 6 km altitude), and so are not very
representative of near-surface $O_3$ concentrations (Cuesta et al., 2018). Due to their small instantaneous field of view and low
Earth orbit, a single satellite instrument is also only capable of providing at best daily observations, while cloudy scenes during
overpasses often means that monthly averaging is required to provide a complete spatial dataset.

To remedy both issues, this work uses data from the Copernicus Atmospheric Monitoring Service (CAMS) reanalysis dataset
(Inness et al., 2019) operated by the European Centre for Medium-Range Weather Forecasts (ECMWF). The CAMS expands
upon the Integrated Forecast System (IFS) used to generate the ERA5 reanalysis (see Section 2.2.2), by assimilating obser-
vations of $O_3$ from a number of instruments. For instance, total columns from nadir-viewing instruments such as the Ozone
Monitoring Instrument (OMI; Levelt et al., 2006) and the Global Ozone Monitoring Experiment-2 (GOME-2; Munro et al.,
2006) provided by the ESA OZONE-CCI dataset (Garane et al., 2018) are assimilated, while stratospheric vertical profiles
are also assimilated from limb-viewing instruments such as the Michelson Interferometer for Passive Atmospheric Sounding
(MIPAS; Fischer et al., 2008) and the Microwave Limb Sounder (MLS; Waters et al., 2006).

Additionally, the CAMS reanalysis also assimilates satellite column observations of several $O_3$ precursor species: tropo-
spheric nitrogen dioxide ($NO_2$), carbon monoxide (CO), and aerosol optical depth (AOD). As well as providing additional
information on tropospheric chemistry, these datasets are also more sensitive to the lower troposphere, and so act to improve
the accuracy of the assimilated near-surface $O_3$ concentrations. The CAMS reanalysis is provided as a 3-hourly dataset with a
$0.75° \times 0.75°$ spatial resolution.

The CAMS reanalysis $O_3$ concentrations over Europe have been validated against surface $O_3$ measurements made by in-
situ stations from the AirBase and European Monitoring and Evaluation Programme (EMEP) monitoring networks (Bennouna
et al., 2020). Most stations reported good agreement with the CAMS reanalysis during summer months ($r > 0.7$). However,
over Southern Europe the CAMS reanalysis was found to consistently overestimate surface $O_3$ concentrations by $\sim 15\%$.
Figure 1 shows the mean surface $O_3$ concentration reported by CAMS for the entire study period.

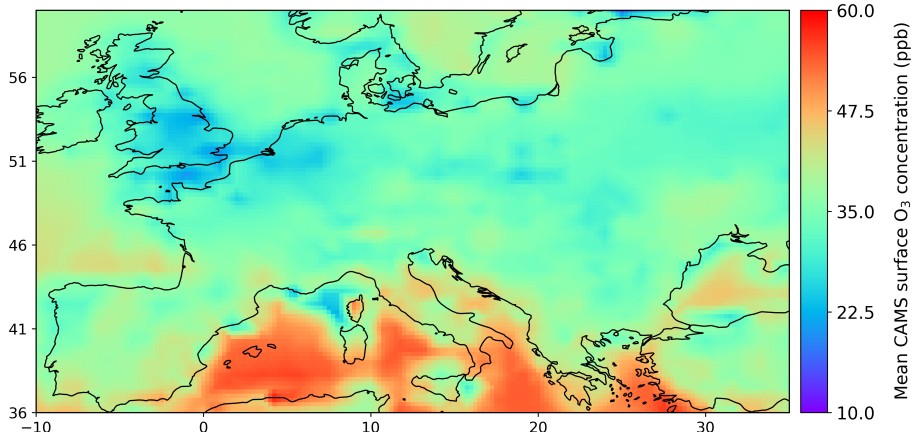

**Figure 1.** Mean surface $O_3$ concentration reported over Europe by the CAMS reanalysis (Inness et al., 2019) between April - September, 2003 - 2015.

### 2.2.2 Meteorology and soil moisture

Computation of the stomatal conductance (see Section 2.3) requires knowledge of the surface air temperature ($T$, °C), vapour pressure deficit ($VPD$, kPa), soil water content ($SWC$, m³m⁻³), and the photosynthetic photon flux density (also known as photosynthetically active radiation, or $PAR$, $\mu$molm⁻²s⁻¹). These variables were all sourced from the ECMWF ERA5 global reanalysis dataset (Hersbach et al., 2020), because of its high spatiotemporal resolution and consistency. ERA5 is produced through 4D-Var assimilation of both in-situ and satellite data to produce a consistent long-term climate dataset. ERA5 is available as an hourly dataset, with a spatial resolution of $0.25° \times 0.25°$.

For this work, the ERA5 2-metre-above-surface temperature was used for $T$, while the $VPD$ was calculated from this and the 2-metre-above-surface dew point temperature. The $PAR$ was calculated from the ERA5 downward solar surface radiation.

ERA5 provides the $SWC$ for four layers of a soil column spanning a depth of 0.0-2.89 m. Comparisons with in-situ measurements suggest very good accuracy (Li et al., 2020), but these measurements do not cover depths below 1.0 m. Therefore, only the top three layers were used for this work (i.e. 0.0-1.0 m).

Information about the soil water content at field capacity ($FC$) and wilting point ($WP$) for these layers was not available, so these variables were instead sourced from the European Soil Data Centre (ESDAC) 3D Soil Hydraulic Database (v1.0; Tóth et al., 2017), which was binned to the ERA5 vertical grid.

### 2.2.3 Land cover

Calculation of stomatal conductance also requires parameters which vary with vegetation type and climate zone. The parameters used in this work are taken from Mills et al. (2017) (see Table 1). Here, forests are classified as either deciduous or
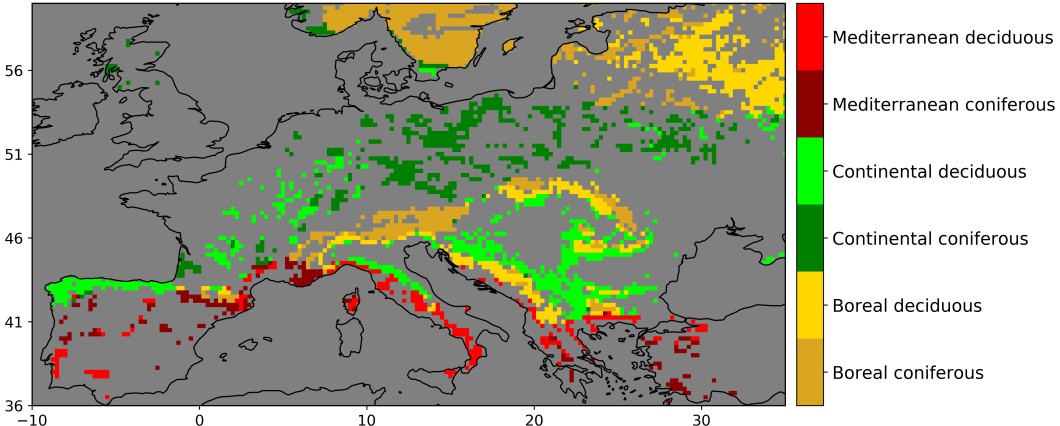

**Figure 2.** Forest type map for 2010 generated from aggregated ESA-CCI land cover (ESA-CCI, 2017) and EEA biogeographical region data (EEA, 2016).

coniferous, and parameters are defined for three biogeographical categories: Boreal (defined here as any mountainous area),
Mediterranean, and Continental. These regions are defined according to the European Environment Agency (EEA) biogeographical regions dataset (EEA, 2016).

Forested regions were identified using annual land cover maps provided by the European Space Agency Climate Change Initiative dataset (ESA-CCI, v 2.0.7; ESA-CCI, 2017), which is a high resolution (300 m) dataset derived from hyperspectral satellite observations. This dataset was aggregated to the ERA5 $0.25° \times 0.25°$ grid using majority resampling. It should be
noted that regions with fragmented forest cover were classified as grassland or cropland, and so were not analysed in this work.

The EEA and ESA-CCI datasets were combined to generate a regional forest type map which was used to determine which parameters to use in the computation of the stomatal conductance. As an example, Figure 2 shows a forest type map for 2010.

### 2.2.4 Phenology

The timings of the start and end of the growing season were calculated from satellite observations of leaf area index (LAI)
from the Global Inventory Modeling and Mapping Studies (GIMMS) LAI3g product (v4, Zhu et al., 2013). This dataset has a 15-day temporal resolution and a $0.083°$ spatial resolution. Estimation of the day of the start and end of the growing season for a given year was performed using the "Four Growing Season Types" (4GST) method discussed in Peano et al. (2019).

### 2.3 Calculation of O$_3$ stomatal conductance

The hourly stomatal conductance to O$_3$ ($g_{sto}$, mmol O$_3$m$^{-2}$s$^{-1}$) was calculated from the ERA5 data discussed in Section 2.2.2
using the Jarvis multiplicative model (Jarvis et al., 1976), which is also used in the DO$_3$SE O$_3$ dry deposition model (Büker et al., 2015) to estimate the risk of O$_3$-induced vegetation damage for European species. In this model, $g_{sto}$ is calculated using





a species-specific function, wherein the maximum possible stomatal conductance, $g_{max}$, is reduced through multiplication by limiting functions, which are scaled between 0-1 based on variables affecting photosynthesis:

$$g_{sto} = g_{max} \cdot f_{phen} \cdot f_{light} \cdot \max(f_{min}, f_T \cdot f_{VPD} \cdot f_{SWC}) \tag{1}$$

where the $f$ terms represent the modification to $g_{max}$ due to phenology ($f_{phen}$), PAR ($f_{light}$), surface air temperature ($f_T$), VPD ($f_{VPD}$), and SWC ($f_{SWC}$). The $f_{min}$ term represents the minimum $g_{sto}$ that occurs during daylight hours, expressed as a fraction of $g_{max}$.

For this work it was asssumed that $f_{phen}$ was 1 throughout the growing season, and otherwise 0. The functions $f_{light}$, $f_T$, and $f_{VPD}$ were calculated using the following formulae, taken from Mills et al. (2017):

$$f_{light} = 1 - \exp\left(-light_a \cdot \text{PAR}\right) \tag{2}$$

$$f_T = \max\left[f_{min}, \left(\frac{T - T_{min}}{T_{opt} - T_{min}}\right) \cdot \left(\frac{T_{max} - T}{T_{max} - T_{opt}}\right)^{\frac{T_{max} - T_{opt}}{T_{opt} - T_{min}}}\right] \tag{3}$$

$$f_{VPD} = \min\left[1, \max\left\{f_{min}, \cdot\frac{(1 - f_{min}) \cdot (VPD_{min} - VPD)}{VPD_{min} - VPD_{max}} + f_{min}\right\}\right] \tag{4}$$

Values for the parameters $f_{min}$, $light_a$, $T_{opt}$, $T_{min}$, $T_{max}$, $VPD_{min}$, and $VPD_{max}$ are species-specific and were taken from Mills et al. (2017) (see Table 1). $f_{SWC}$ was calculated using the improved function described in Anav et al. (2018):

$$f_{SWC} = \min\left[1, \max\left\{f_{min}, \frac{SWC - WP}{FC - WP}\right\}\right] \tag{5}$$

Plants are known to adsorb water from soil depths with the highest water availability (Anav et al., 2018). Therefore, the mean $SWC$ over the top three ERA5 soil layers at a given time was used to compute $f_{SWC}$, along with the mean $FC$ and $WP$ for these layers.

Figure 3 shows the mean hourly $g_{sto}$ calculated for July 2010 as an example of this computation. During the summer
months, both coniferous and deciduous forest $g_{sto}$ appears to be strongly modulated by spatial variations in temperature and soil moisture.



**Table 1.** The species-specific parameters from Mills et al. (2017) used in Equations 1-5 to compute the stomatal conductance to $O_3$ ($g_{sto}$). The $T_{max}$ value for Boreal trees is set to 200°C in order to simulate the weak temperature response of trees growing in Northern European conditions.

| Region | Units | Boreal | | Continental | | Mediterranean | |
|---|---|---|---|---|---|---|---|
| Forest type | | Deciduous | Coniferous | Deciduous | Coniferous | Deciduous | Evergreen |
| $g_{max}$ | mmol $O_3 m^{-2} s^{-1}$ | 240 | 125 | 155 | 130 | 265 | 195 |
| $f_{min}$ | fraction | 0.1 | 0.1 | 0.13 | 0.16 | 0.13 | 0.02 |
| $light_a$ | - | 0.0042 | 0.006 | 0.006 | 0.01 | 0.006 | 0.012 |
| $T_{min}$ | °C | 5 | 0 | 5 | 0 | 1 | 0 |
| $T_{opt}$ | °C | 20 | 20 | 16 | 14 | 22 | 23 |
| $T_{max}$ | °C | 200 | 200 | 33 | 35 | 35 | 39 |
| $VPD_{min}$ | kPa | 0.5 | 0.8 | 1.0 | 0.5 | 1.1 | 2.2 |
| $VPD_{max}$ | kPa | 2.7 | 2.8 | 3.1 | 3.0 | 3.1 | 4.0 |

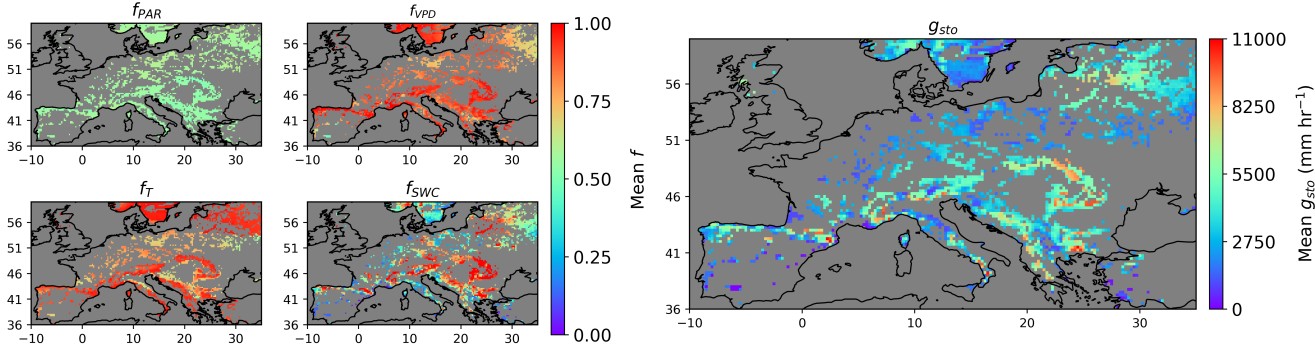

**Figure 3.** Mean hourly stomatal conductance to $O_3$ ($g_{sto}$) calculated using ERA5 data and parameters listed in Table 1 for July 2010. Left: The mean $f$ terms calculated using Equations 2-5, Right: $g_{sto}$ calculated using Equation 1.

## 2.4 Estimation of GPP reduction due to $O_3$

The GPP reduction due to $O_3$ exposure is estimated from $g_{sto}$ and CAMS $O_3$ concentrations using the method established previously in Anav et al. (2011) and Proietti et al. (2016). The instantaneous GPP reduction due to $O_3$ exposure ($I_{O_3}$) can be expressed as a dimensionless fraction of the GPP without $O_3$ damage using the following relation:

$$I_{O_3} = \alpha \cdot g_{sto} \cdot AOT40 \tag{6}$$





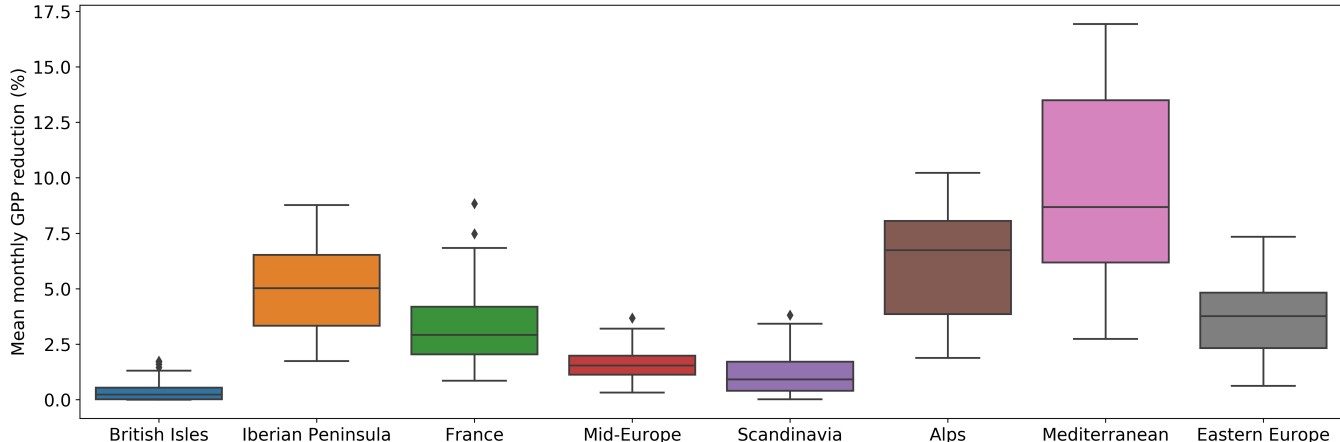

**Figure 4.** Box and whisker plot of the monthly mean estimated $O_3$-induced GPP reduction over each European climatic region defined in (Christensen and Christensen, 2007). The box indicates the value of the median and interquartile range (IQR) between the 25th (Q1) and 75th (Q2) percentiles, while the whiskers show the range from Q1-1.5*IQR to Q3+1.5*IQR. Outliers are indicated with dots.

where $AOT40$ is the hourly Accumulated $O_3$ exposure Over a Threshold of 40 ppb (ppb h), and $\alpha$ is an empirically derived $O_3$ response coefficient ($mm^{-1}ppb^{-1}$). If $g_{sto}$ is converted to the units mm $hr^{-1}$ (See Mills et al. (2017)), then $I_{O_3}$ becomes dimensionless. For coniferous forests $\alpha$ was set to $0.7\times10^{-6}$ (Reich, 1987), while for deciduous forests an $\alpha$ value of $2.6\times10^{-6}$
was used (Ollinger et al., 1997).

## 3    Results

### 3.1    $O_3$-induced GPP reductions over Europe

The maximum, minimum, and mean monthly $O_3$-reduction calculated using Equation 6 for the entire analysis period is shown in Figure 5, while Figure 4 shows the temporal range of the spatial mean for each European climatic region defined in Christensen and Christensen (2007).

Figure 5 shows that across Europe the mean GPP reduction ranged from 0.1%-25%. From Figures 5 and 4 it is also clear that estimated GPP reductions exhibit high temporal variability; over Italy the estimated GPP reductions exceed 50% for some months, while over Northeastern Europe the estimated reduction is almost 0% for other months.

$O_3$ concentrations and $g_{sto}$ over the Mediterranean are higher than in Northern Europe, which results in the negative latitudi-
nal gradient between Southern and Northern European GPP reductions. Coniferous forests exhibit much lower GPP reductions than deciduous forests over all regions, as they are more resistant to $O_3$ exposure.







**Figure 5.** The minimum, maximum, and mean monthly $O_3$-induced GPP reduction ($I_{O_3}$) between April-September, 2003-2015, calculated from satellite data for European forests using Equation 6.

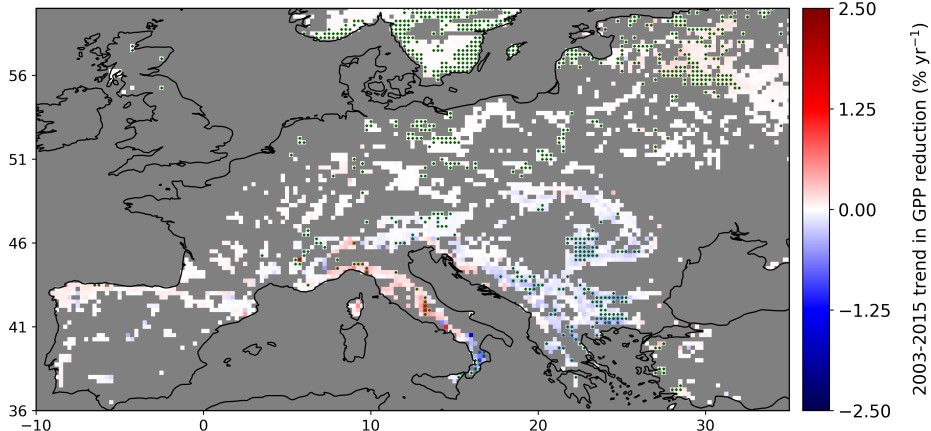

**Figure 6.** The annual trends in the empirically estimated $O_3$-induced GPP reductions between 2003-2015. Statistically significant ($p < 0.05$) trends are indicated with green dots.

## 3.2 Trend analysis

Temporal variability in the estimated GPP reductions was analysed by computing the trend in the annual mean reductions for each grid cell using linear regression. The calculated trends are shown in Figure 6.

While most regions do not exhibit a statistically significant trend, over Scandinavia and Eastern Europe there are statistically significant ($p < 0.05$) trends of approximately +0.5% $yr^{-1}$, while over Greece and the Balkans the GPP reductions appear to decline by approximately -0.5% $yr^{-1}$.

Over Italy, the GPP reductions appear to be increasing by up to +1.0% $yr^{-1}$ over most regions excluding Calabria (lat: 39.00°, lon: 16.45°), over which the GPP reductions appear to be decreasing by a similar rate. However, these trends are statistically insignificant for almost all grid cells in this region.

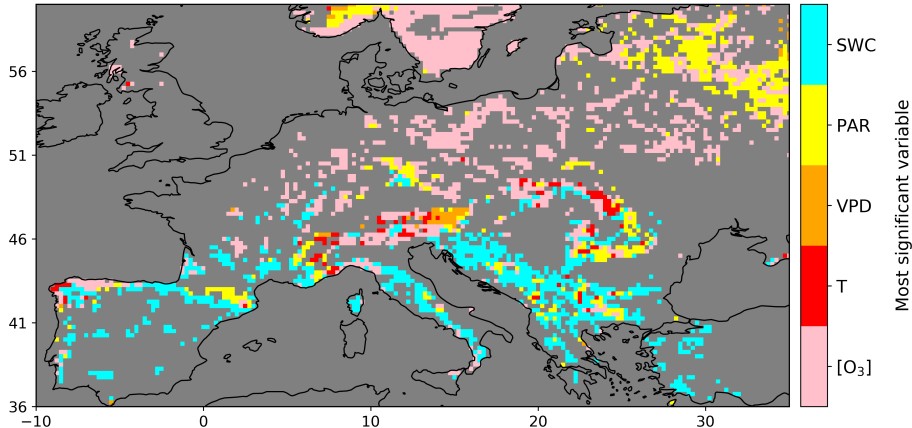

**Figure 7.** The most important variable affecting O₃-induced GPP reductions estimated in Figure 5, determined using Random Forest analysis (RFA); T: Temperature, SWC: Soil water content, PAR: Photosynthetically active radiation, VPD: Vapour pressure deficit, [O₃]: O₃ concentration.

## 3.3 Random Forest analysis

Determining the cause of the spatiotemporal variability in the GPP reductions calculated using Equation 6 is non-trivial, as trends in both O₃ concentration and the climatic variables which contribute to $g_{sto}$ (VPD, temperature, SWC, and PAR) must all be considered.

Following Proietti et al. (2016), the estimated GPP reductions were regressed against the O₃ concentration and the ERA5 climatic variables using Random Forest analysis (RFA; Breiman, 2001). This is a machine learning technique, in which an ensemble of decision trees are grown in a randomly selected subspaces of a given dataset. RFA has previously been found to be robust and capable of accounting for correlation and interaction effects among variables, and so has previously been used to disentangle the effects of O₃ and climatic factors on vegetation (Proietti et al., 2016; Sicard and Dalstein-Richier, 2015). The

normalised Gini importance from the fit was then used to identify the most important variable.

    Figure 7 shows the most important variable driving the estimated GPP reduction for each grid cell. For most of the Mediterranean region, soil water content appears to be the most significant variable limiting GPP reductions, while O₃ concentration appears to be the most significant variable for most of central Europe. PAR also appears to be the most significant variable over Northeastern Europe, while temperature and VPD appear to be the most significant variable over high terrain such as the Alps

and Carpathian mountains.





## 3.4 Comparison with Yale Interactive Biosphere (YIBs) model simulations

Validation of these estimates was performed through direct comparison with $O_3$-induced GPP reductions between 2003-2011 simulated by the Yale Interactive Biosphere model (YIBs; Yue and Unger, 2015). These estimates were previously simulated as part of the Yue and Unger (2018) study investigating the effect of fire emissions on global GPP, though this particular dataset

does not include fire-induced emissions.

In this dataset GPP over European forests was simulated with and without $O_3$ exposure, while $O_3$-induced GPP reductions were calculated using low and high vegetation sensitivity scenarios defined in Sitch et al. (2007). $O_3$-induced GPP reductions can therefore be expressed as a percentage of the $O_3$-free GPP. For this comparison, the satellite-based estimates were regridded to the $1° × 1°$ spatial grid used by YIBs, and only grid cells where both YIBs and the satellite data agreed were forested were

compared.

Figure 8 shows that the satellite-based estimates agree better with the low $O_3$ sensitivity simulations, as shown by the lower overall magnitude of the bias. However, the satellite-based GPP reductions are $\sim 12\%$ (absolute value) greater than YIBs over the Mediterranean.

This bias is spatially and temporally consistent over the entire analysis period, suggesting a fundamental difference between

the assumptions used in either method. This difference could be caused by differences in the definition of soil water stress to vegetation, which is likely better accounted for in a process-based model like YIBs, instead of the simpler Jarvis multiplicative algorithm used in this work.

## 4   Discussion and Conclusions

Maintenance and restoration of terrestrial ecosystems are vital to climate change mitigation, but current strategies do not take

into account the indirect effects of pollution on vegetation. Exposure to anthropogenic $O_3$ is a particular concern, as it is known to damage vegetation and so directly inhibit photosynthesis. This effect in turn may jeopardise attempts to mitigate dangerous climate change by further degrading terrestrial carbon sinks.

Prior attempts to quantify this effect have relied on sparse in-situ measurements or land surface models. Satellite observations offer long-term synoptic coverage of atmospheric composition, meteorology, and vegetation properties. This work is the first

attempt to estimate the effect of $O_3$ on the European carbon sink using primarily satellite data.

In this work $O_3$-induced GPP reductions were estimated using the linear empirical method previously defined in Anav et al. (2011) and Proietti et al. (2016). Stomatal conductance ($g_{sto}$) was estimated for European forests using ERA5 reanalysis data, and then combined with CAMS reanalysis $O_3$ concentrations and literature response functions to estimate the relative reduction in photosynthesis.

As shown in Figure 5, there is a clear latitudinal gradient in the mean estimated GPP reductions over Europe, caused by differences in parameters defined for coniferous and deciduous vegetation. The mean estimated GPP reductions vary between 0.36% over the British Isles to 9.55% over the Mediterranean. However, the GPP reductions approach almost 50% over Italy for certain months during high $O_3$ concentration events.





Figure 8. Annual mean monthly bias (April-September) between $O_3$-induced European forest GPP reductions estimated using satellite data and the YIBs model (Yue and Unger, 2018). YIBs simulations are based on low (a) and high (b) vegetation sensitivity to $O_3$ as defined in Sitch et al. (2007). The satellite-based estimates were regridded to the $1° \times 1°$ YIBs spatial resolution for this comparison.





Prior investigations into European $O_3$-induced GPP reductions reported comparable results to this work. Anav et al. (2011)
used the ORCHIDEE land surface model to estimate a 22% total reduction in European GPP during 2002, nearing 45% over
some locations. While this is much larger than the estimates derived in this work, it must be noted that these estimates also
include crops and grassland, which are much more sensitive to $O_3$ exposure than trees (Reich, 1987). More recently, Oliver et al.
(2018) estimated GPP reductions of 2-8% for European boreal regions and 10-20% for temperate regions between 1901-2050,
using the JULES land surface model. Again, these estimates are positively biased by their inclusion of crop and grassland.

The $O_3$-induced GPP reductions estimated in this work are also similar to those estimated by Proietti et al. (2016) for
European forest sites between 2000-2010 using the same empirical method. In that study, the mean annual $O_3$-induced GPP
reductions were found to vary between 0-12% along a similar latitudinal gradient, but could reach up to 30% over Switzerland
and Slovakia. $O_3$ concentrations in the Proietti et al. (2016) study were directly sourced from in-situ measurements, while
the parameters used in Equation 1 were taken from the precursor to the ERA5 reanalysis dataset, ERA-Interim (Dee et al.,
2011). As $O_3$ concentrations predicted by CAMS agree well with in-situ measurements (Bennouna et al., 2020), it is therefore
expected that these estimates would agree.

Figure 6 shows that small yet statistically significant trends in the annual estimated GPP reductions exist over Scandinavia
(+0.05 % yr$^{-1}$), the Balkans (-0.5 % yr$^{-1}$), and Eastern Europe (+0.5% yr$^{-1}$). Over Italy there is a stronger but statistically
insignificant positive trend of $\sim$1.0 % yr$^{-1}$ over most regions except the far South, where the trend becomes $\sim$-1.0 % yr$^{-1}$.

The trends in GPP reduction estimated in this work are dependent on both European $O_3$ concentrations and the climatic
variables governing stomatal conductance. By modelling both changes in ground-level $O_3$ concentration and dry deposition,
Proietti et al. (2020) estimated statistically significant decreases in annual $O_3$ concentration (-2%) and AOT40 (-26.5%) across
Europe between 2000-2014, indicating the large-scale success of European pollutant control strategies. However, the same
time period also saw significant increases in Phytotoxic $O_3$ Dose (POD) across European forests, particularly after 2010. This
suggests that damage to vegetation may have actually increased, primarily due to climate change increasing the growing season
length (Anav et al., 2019). Additionally, positive changes in temperature and PAR would also increase $g_{sto}$, and therefore
increase the stomatal flux of $O_3$, even as the average concentration decreases.

RFA was used to determine the main factors driving the spatiotemporal variability shown in Figures 5 and 6. From Figure 7
it is clear that soil moisture is a significant variable governing the estimated GPP reductions over the Mediterranean. A similar
result was found by Anav et al. (2018), who determined that soil water availability significantly limited stomatal conductance
and so $O_3$ dry deposition in semi-arid regions. Elsewhere, the influence of $O_3$ concentrations on the estimated GPP reductions
was found to be the most important over most of central and Northern Europe. However, over mountainous regions the effect
of temperature, PAR, or VPD were found to be the most significant. A possible explanation may be that these regions are
experiencing much faster warming than lower terrain (Pepin et al., 2015), and so the effect of these variables on growing
season and stomatal conductance would be much greater than the relatively declining $O_3$ concentration.

The GPP reduction estimates determined in this work were validated against $O_3$-induced GPP reductions estimated between
2003-2011 using the YIBs land surface model by Yue and Unger (2018). The satellite-based GPP reduction estimates were
found to agree well with reductions predicted by the YIBs low vegetation $O_3$ sensitivity scenario. However, the satellite-based





reductions exhibited a consistent +12% bias against the YIBs estimates over the Mediterranean for all years, indicating that

there is a fundamental difference in either how $g_{sto}$ or $O_3$ sensitivity is computed between YIBs and this work. One possible explanation is that YIBs uses the Ball-Berry model (Ball et al., 1987) to compute $g_{sto}$ instead of the Jarvis model used in this work.

Another important factor to consider is that YIBs has a coupled photosynthesis-$g_{sto}$ scheme, such that vegetation damaged by $O_3$ have decreased $g_{sto}$ (Yue and Unger, 2015). This reflects the fact that damaged stomatal cells are inclined to remain

closed due to changes in cell turgor pressure and signalling pathways. As a result, there is an upper limit to the modelled $O_3$-induced GPP reductions. By contrast, Equation 6 assumes a simple linear dependence between GPP and $g_{sto}$, and so offers no limitation of $g_{sto}$ with $O_3$ damage. However, there is evidence to suggest that prolonged $O_3$ exposure may cause photosynthesis and $g_{sto}$ to decouple over time, thereby removing this limiting factor (Lombardozzi et al., 2013). Further comparisons against field observations are necessary to determine to what extent this simplification affects the accuracy of the predicted GPP

reductions.

The applicability of the satellite-based model used in this work may be limited by the fact that Equation 6 uses AOT40 as a measure of $O_3$ exposure, and so assumes that vegetation damage can occur after any deposition of $O_3$. However, in the $DO_3SE$ model vegetation exposure to $O_3$ is modelled using the Phytotoxic $O_3$ Dose above a threshold Y (PODY) metric. This formulation assumes that below a given threshold flux, vegetation can safely detoxify absorbed $O_3$ without any injury (Mills

et al., 2011). Currently, a threshold value of 1 nmol m$^{-2}$ s$^{-1}$ is recommended for all trees by Mills et al. (2017). That said, it has also been suggested that no threshold $O_3$ flux should be assumed when estimating vegetation damage, as the detoxification processes are dynamic and so cannot be represented by a single constant (Musselman et al., 2006). Again, comparisons with field measurements are needed to determine whether the model could be improved through the addition of a threshold exposure.

It should be noted that the spatiotemporal resolution and accuracy of satellite data will only improve as the next generation

of instruments come online. For instance, geostationary missions such as Sentinel-4 (Gulde et al., 2017) and the Tropospheric Emissions: Monitoring of Pollution (TEMPO; Zoogman et al., 2017) will provide hourly daytime measurements of both tropospheric $O_3$ and its precursors for the first time, potentially improving near-surface $O_3$ concentrations derived from reanalyses like CAMS. Additionally, satellite observations of solar-induced fluorescence (SIF) have recently offered a functional proxy of photosynthesis, and so GPP (e.g. Magney et al., 2019). Future studies using the satellite-based approach discussed in this

work would therefore be improved by the inclusion of such datasets.

Overall, this work has demonstrated for the first time that satellite-based datasets can be leveraged to produce estimates of $O_3$-induced GPP reductions, which are comparable with in-situ or model-based analyses. The spatial features and temporal trends of the GPP reductions reported in this work support the conclusions of previous investigations, that $O_3$ exposure remains a significant risk to the European carbon sink, despite the success of emission reduction policies. The results of the RFA also

underline the importance of regional climate on modulating or even enhancing the impact of $O_3$ on photosynthesis, particularly over the Mediterranean. The satellite-based approach described in this work could potentially be applied to investigate currently poorly sampled regions such as the Amazon to better quantify the effect of $O_3$ on the global terrestrial carbon sink, at a fraction of the computational cost otherwise required by land surface models.



*Code availability.* The YIBs model is available from: https://github.com/YIBS01/YIBS_site.

The 4GST code used to calculate the vegetation phenology is available from: https://github.com/daniele-peano/4GST.

*Data availability.* All datasets used in this work have been made freely available by their authors.

The CAMS reanalysis dataset can be downloaded from the Copernicus Atmosphere Data Store (https://ads.atmosphere.copernicus.eu/!/home, last access: 09 May 2021), while the ERA5 reanalysis dataset can be downloaded from the Copernicus Climate Data Store (https://cds.climate.copernicus.eu/#!/home, last access: 09 May 2021).

ESA-CCI Land Cover data can be downloaded from http://maps.elie.ucl.ac.be/CCI/viewer/download.php (last access: 09 May 2021).

The GIMMS LAI3g dataset are available on request from Zaichun Zhu (zhu.zaichun@pku.edu.cn)

*Author contributions.* JSA designed the study, performed the analysis, and wrote the manuscript. AA developed the empirical model for calculating $O_3$-induced GPP reductions. Both AA and MV assisted in interpreting the modelled stomatal conductance and estimated GPP reductions. DP provided the 4GST code and helped to adapt it for this work. NU and XY provided the YIBs data and provided context to the

observed differences between the YIBs and satellite-based results. RJP and HB recommended analyses and helped to draft the manuscript. All authors provided comments and improvements to the manuscript.

*Competing interests.* The authors declare that they have no conflict of interests.

*Acknowledgements.* JSA was funded by the ESA Living Planet Fellowship, EOCYTES: Evaluation of the effect of Ozone on Crop Yields and the TErrestrial carbon pool using Satellite data, 2019-2020 (4000125584/18/I-NS)

RJP, HB and JSA are funded via the UK National Centre for Earth Observation (NE/N018079/1 and NE/R016518/1).

This research used the ALICE High Performance Computing Facility at the University of Leicester, and JASMIN, the UK's collaborative data analysis environment (http://jasmin.ac.uk; Lawrence et al., 2013)



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
