# Peer review of "Ozone-induced gross primary productivity reductions over European forests inferred from satellite observations"

_Biogeosciences, 2021_

## Referee Comment (RC1)

**Review of bg-2021-125, by Singh Anand et al. on "Ozone-induced gross primary productivity reductions over European forests inferred from satellite observations"**

The paper presents an analysis of the impact of ozone versus meteorological drivers and soil moisture on European forest's GPP. It claims to be the first study to evaluate this impact at the continental scale using satellite observations. I deem this claim to be not being supported by the presented approach. There is indeed use of some satellite data in the form of the LAI and forest cover data (and you can argue that the CAMS $O_3$ data partly rely on the assimilation of remote sensing data). Having read initially the abstract I anticipated that the LAI timeseries were going to be used as a proxy for changes in GPP. But the crucial component of this study, GPP does not rely at all on the use of any source of satellite observations. The impact of $O_3$ on GPP is calculated in the presented study as a model product using some empirical constants, stomatal conductance and the accumulated $O_3$ concentrations. As such the presented analysis can mostly be interpreted as a validation step of the followed approach integrating the spatio-temporal information of many different datasets of relevant parameters. This is then complemented with a sensitivity analysis to indicate the spatio-temporal patterns in the role of $O_3$ vs the meteorology drivers of GPP. In addition, then comparing then the results of the ERA5/CAMS/etc. based $O_3$-GPP model with another model that also includes some empirical relationships to consider the $O_3$ impact on simulated GPP is mainly another validation step, e.g., showing that the model(s) is/are properly implemented. The main shortcoming of this paper is that there is not specific evaluation step; optimally one would have applied remote sensing based vegetation indices/GPP estimates. One already missed opportunity for some first evaluation step of the followed approach would have been evaluation of the inferred stomatal conductances, e.g., comparing the Jarvis based latent heat flux with FLUXNET observations. Consequently, quantification of the large-scale $O_3$ impact on vegetation functioning using large-scale satellite observations of vegetation dynamics, as all suggested by the title, abstract and introduction, is according to me not addressed. Based on these observations and considerations, I recommend the paper to be rejected.

Despite this overall negative recommendation, I share here also the specific comments that came up reading the paper and which can potentially be used for future revisions.

**Specific comments:**

Pp 1, lines 16-18: A bit confusing the first number in these lines (i.e., 30%) refers to the net uptake of $CO_2$ by vegetation (NEE, not GPP), and the carbon fluxes that are quoted after that are in fact GPP.

Pp 3, line 55 "Similarly, eddy covariance towers, such as those that make up the global FLUXNET dataset (Pastorello et al., 2020) can also be employed to investigate the effect of $O_3$ exposure on GPP". Here you could add the reference(s) to the work by Ducker et al., Biogeosciences, 15, 5395–5413, 2018, https://doi.org/10.5194/bg-15-5395-2018

Line 64: "soil hydrology" which soil hydrological variable? This is not specific.

Pp 3, line 73: "near-surface $O_3$ concentration and meteorology governing GPP", using this statement expresses that you deem that GPP is controlled by meteorological drivers and $O_3$ (concentrations). But what are the parameters all known to effect GPP; I missing here in the introduction a mentioning of other parameters that might be important and that might not be easily inferred from remote sensing data, e.g. N-deposition.

Pp 4, line 84, "Excluding soil moisture and meteorology", I don't get this statement; You refer to the method of regridding. Do you mean here that for all other parameters than soil moisture and meteorology you have applied this regridding procedure. But then mentioning here the term soil moisture, it would be good to already indicate in the introduction how this parameter can play a (crucial) role in inferring the $O_3$ impact on GPP.

Pp 4, lines 113-115; Reading the statements about to what extent the CAMS $O_3$ can be applied for assessing its impact on GPP, just giving the overall statistics expressed by this r value of 0.7 ($r^2 < 0.5$, is actually not such a high-correlation) triggers the question if this applies for summer mean/monthly/diurnal mean/max, or full timeseries? This is of large relevance since what matters most for this impact assessment is how well CAMS captures the high $O_3$ (extremes) during the days when stomatal uptake is maximum. In addition, you state: "However, over Southern Europe the CAMS reanalysis was found to consistently overestimate surface $O_3$ concentrations by ~15%.". Other terms in Eq. 6 (e.g., the alpha-term) can have considerable uncertainty. This uncertainty should be propagated in the GPP reduction estimate, especially considering that your model only requires "a fraction of the computational cost otherwise required by land surface models" (Lines 317-318).

Pp 4, lines 126: this motivation of only using the soil moisture of the top layers excluding the information on soil moisture > 1m indicates that you assume that the forests stomatal conductance is mainly controlled by the soil moisture in the top 1m. This might actually depend a lot on the effective rooting depth. I bring this up having seen soil moisture observations in the top soil profiles that seemed to provide a nice source of information to indeed infer the impact of soil water on stomatal conductance but where, then evaluating the observed latent heat fluxes, did not not reflect at all observed strong decreases in those soil moisture measurements.

Pp 9: lines 175-180; At the end of the methods having seen the overview of all the datasets being used, it makes me wonder about any evaluation strategy that you have developed to at least assess that some of the critical parameters in your inversion of the $O_3$ impact make sense; e.g., did you conduct any evaluation of the Jarvis stomatal conductance based on comparison of the simulated and observed LE? This is a parameter that could have been rather easily evaluated using the FLUXNET datasets.

And why using the AOT40 where in the previous paragraphs you have referred to the use of Jarvis in the DoseO3 model to evaluate the stomatal dose of $O_3$? This has to be all better motivated and including the potential implications.

Figs. 4 and 5: These figures are derived from monthly $O_3$-induced GPP estimates. It would be interesting to show the actual monthly data over the growing season (i.e. a

time series), this would provide more insight in the dynamics. The boxplot in Fig. 4 is in fact a bit misleading, as this is typically used to show represent errors/uncertainties, but a proper error propagation is lacking

---

## Author Comment (AC1)

We thank the reviewer for their comments, and will respond to them herein.

**The paper presents an analysis of the impact of ozone versus meteorological drivers and soil moisture on European forest's GPP. It claims to be the first study to evaluate this impact at the continental scale using satellite observations. I deem this claim to be not being supported by the presented approach. There is indeed use of some satellite data in the form of the LAI and forest cover data (and you can argue that the CAMS $O_3$ data partly rely on the assimilation of remote sensing data).**

Please allow us to elaborate on why we believe that the claim is appropriate. All parameters in this work relevant to stomatal conductance and ozone exposure have either been entirely sourced from satellite datasets (e.g. phenology, forest cover) or from reanalysis datasets which heavily rely on assimilated satellite observations (e.g. $O_3$, meteorology). To our knowledge, no prior work has entirely relied on satellite data in this way.

In the case of the reanalysis datasets, validation studies against ground-based observations for ERA5 (Hersbach et al, 2020) and CAMS (Inness et al, 2019; Huijnen et al, 2020) indicate that they offer superior accuracy compared to models which do not assimilate satellite data. As such, the regional-scale analyses discussed in this work would have been impossible to perform using other observation-based datasets. This discussion will be added to the final draft of the paper.

**But the crucial component of this study, GPP does not rely at all on the use of any source of satellite observations. The impact of $O_3$ on GPP is calculated in the presented study as a model product using some empirical constants, stomatal conductance and the accumulated $O_3$ concentrations. As such the presented analysis can mostly be interpreted as a validation step of the followed approach integrating the spatio-temporal information of many different datasets of relevant parameters.**

As with terrestrial FLUXNET measurements, satellite-based observations of GPP are of vegetation that have already been exposed to ambient $O_3$ concentrations, so quantifying the GPP lost due to $O_3$ exposure is not possible without first estimating the what the vegetation GPP would be in the absence of any $O_3$. Additionally, satellite GPP datasets are not direct observations, but are instead inferred from other measurements using models as well (e.g. MODIS MOD17; Running and Zhao, 2015). Therefore, analyses of such datasets would also need to determine and distinguish biases introduced by these models from genuine changes due to $O_3$.

Our method circumvents these issues by focusing only on the relative change in GPP expected based on variables known to affect the vegetative $O_3$ flux, and using a stomatal conductance model that has already been well validated in the literature. We will add this justification to the final paper draft.

**The main shortcoming of this paper is that there is not specific evaluation step; optimally one would have applied remote sensing based vegetation indices/GPP estimates. One already missed opportunity for some first evaluation step of the followed approach would have been evaluation of the inferred stomatal conductances, e.g., comparing the Jarvis based latent heat flux with FLUXNET observations.**

We agree that an explicit evaluation step is missing from the current manuscript, and will now discuss herein such a comparison with FLUXNET observations we performed in response to these comments, which will be added to the final paper draft.

We identified 11 FLUXNET sites (see Table 1) which were suitable for comparison with our data (i.e. situated in forests and recorded observations between 2003 – 2015, according to the FLUXNET2015 dataset; Pastorello et al, 2015). These sites are shown in Table 1 and Figure 1.

For this exercise, we compared our satellite-based stomatal conductances with the monthly averaged values estimated from FLUXNET latent heat measurements by Ducker et al (2018) as part of the SynFlux dataset. The FLUXNET relative humidity and precipitation measurements were used to perform the same filtering of the satellite-based stomatal conductance data as in Ducker et al (2018); only stomatal conductances reported during daylight hours, when the relative humidity was < 80%, and for days with < 5 mm precipitation were used to compute the monthly averages.

The satellite-based stomatal conductances estimated using the Jarvis model are modelled per leaf unit area, so they must be scaled with an appropriate leaf area index (LAI) in order to compare these against the SynFlux values. To do this, we used the Global Monthly Mean Leaf Area Index Climatology derived from the GIMMS LAI3g product (Mao and Yan, 2019).

Several statistics were calculated for each FLUXNET site to assess the accuracy of the satellite-based stomatal conductances against the SynFlux dataset. In addition to the monthly mean bias and coefficient of determination ($R^2$), the satellite-SynFlux bias was quantified using the Modified Normalised Mean Bias (MNMB) and Fractional Gross Error (FGE):

$$\text{MNMB} = \frac{2}{N} \sum_i^N \frac{o_i - s_i}{o_i + s_i} \quad [1]$$

$$\text{FGE} = \frac{2}{N} \sum_i^N \left| \frac{o_i - s_i}{o_i + s_i} \right| \quad [2]$$

Here, $o_i$ and $s_i$ are the mean values for a given month $i$ calculated from the satellite-based and SynFlux stomatal conductance, respectively, and $N$ is the sample size. The MNMB and FGE statistics scale all biases in a symmetric range between -2 – 2, and 0 – 2, respectively. Both statistics also maintain a linear estimation of the bias, and so are less sensitive to extreme outliers compared to raw means or root mean squares.

Table 1 and Figure 1 show the results of this comparison. As shown in Figure 1, the 11 available FLUXNET sites occupy a longitudinal band between 6° – 14° E, so it was not possible to validate stomatal conductances for forests outside of central Europe and Italy. Similarly, none of the sites were located in Mediterranean deciduous forests, so no validation was possible for this vegetation type.

It is clear that with the exception of DK-Sor (the only deciduous forest site), there are significant differences between the SynFlux and satellite-based stomatal conductance values for all FLUXNET sites. The stomatal conductance biases shown in Table 1 may be a result of the ERA5 data used in the Jarvis model.

To investigate this further the air temperature, VPD, shortwave radiation flux, and soil water content observed at each FLUXNET site during the same time period were compared with the corresponding value reported by ERA5. Table 2 shows the MNMB, FGE, and $R^2$ calculated from the monthly mean values. While temperature, VPD, and shortwave flux show good agreement with the in-situ data over all sites, the ERA5 soil water content correlates very poorly with the FLUXNET measurements.

Though soil water content is not used to explicitly calculate the SynFlux stomatal conductance, it is nevertheless a vital parameter regulating stomatal conductance (Anav et al, 2018). It is likely that this disagreement is the reason for the observed stomatal conductance biases, so any future work will need to test other satellite-based root-zone soil moisture products (e.g. H-SAF, 2020) to determine if they are more suitable than ERA5 for estimating stomatal conductance.

**Pp 1, lines 16-18: A bit confusing the first number in these lines (i.e., 30%) refers to the net uptake of $CO_2$ by vegetation (NEE, not GPP), and the carbon fluxes that are quoted after that are in fact GPP.**

We agree that the first sentence is potentially confusing, and will remove the citation for NEE in the final draft of this paper.

**Pp 3, line 55 "Similarly, eddy covariance towers, such as those that make up the global FLUXNET dataset (Pastorello et al., 2020) can also be employed to investigate the effect of $O_3$ exposure on GPP". Here you could add the reference(s) to the work by Ducker et al., Biogeosciences, 15, 5395–5413, 2018, https://doi.org/10.5194/bg-15-5395-2018**

The SynFlux $O_3$ deposition dataset was previously referred to in the FLUXNET comparison exercise discussed above, and will be added to the final draft.

**Line 64: "soil hydrology" which soil hydrological variable? This is not specific.**

This was meant to reference the satellite datasets for soil moisture, namely the volumetric soil water content provided by ESA-CCI. We will change this line in the final draft.

**Pp 3, line 73: "near-surface $O_3$ concentration and meteorology governing GPP", using this statement expresses that you deem that GPP is controlled by meteorological drivers and $O_3$ (concentrations). But what are the parameters all known to effect GPP; I missing here in the introduction a mentioning of other parameters that might be important and that might not be easily inferred from remote sensing data, e.g. N deposition.**

The reviewer is correct to point out that meteorology and $O_3$ are not the only parameters known to affect GPP, and we will certainly add a discussion on these parameters in the final draft. However, in many existing vegetation models (e.g. Lombardozzi et al, 2015) the reduction in photosynthetic rate (and so GPP) due to $O_3$ exposure is solely dependent on the cumulative $O_3$ uptake, which in turn is a function of $O_3$ concentration and stomatal conductance. Therefore, focusing on only $O_3$ and meteorological parameters was justified for this work. We will add this point to the final paper draft.

**Pp 4, line 84, "Excluding soil moisture and meteorology", I don't get this statement; You refer to the method of regridding. Do you mean here that for all other parameters than soil moisture and meteorology you have applied this regridding procedure. But then mentioning here the term soil moisture, it would be good to already indicate in the introduction how this parameter can play a (crucial) role in inferring the $O_3$ impact on GPP.**

The sentence was meant to state that all datasets were regridded to the ERA5 spatial grid (i.e. the source of the meteorological and soil moisture data). We will make this clearer and add information to the final draft introduction section about the influence of soil moisture on stomatal conductance and the subsequent impact on the vegetation $O_3$ flux.

**Pp 4, lines 113-115; Reading the statements about to what extent the CAMS $O_3$ can be applied for assessing its impact on GPP, just giving the overall statistics expressed by this r value of 0.7 ($r^2 < 0.5$, is actually not such a high-correlation) triggers the question if this applies for summer**

**mean/monthly/diurnal mean/max, or full timeseries? This is of large relevance since what matters most for this impact assessment is how well CAMS captures the high $O_3$ (extremes) during the days when stomatal uptake is maximum. In addition, you state: "However, over Southern Europe the CAMS reanalysis was found to consistently overestimate surface $O_3$ concentrations by ~15%.". Other terms in Eq. 6 (e.g., the alpha-term) can have considerable uncertainty. This uncertainty should be propagated in the GPP reduction estimate, especially considering that your model only requires "a fraction of the computational cost otherwise required by land surface models" (Lines 317-318).**

The r > 0.7 for most stations reported by Bennouna et al (2020) was calculated for averaged monthly data between April-September. It should be noted that since the original submission of this paper, Wagner et al (2021) published a more detailed comparison of CAMS and in-situ $O_3$. They found that while CAMS reproduces tropospheric $O_3$ within 10% of independent observations, a seasonal mean variability in biases exist over Northern midlatitudes, peaking at 15% in October.

Over Northern Europe, and to a lesser extent Southern Europe, it was found that the CAMS overestimation of surface $O_3$ was greater during nighttime than daytime hours (roughly twice as large than daytime biases). They propose that this indicates that nocturnal $O_3$ destruction processes in the boundary layer (e.g. $NO_x$ titration) are not being correctly included in the global model, which is known to have difficulties resolving subgrid processes. We will add this reference and explanation in the final paper draft.

Regarding error analyses, the above FLUXNET comparisons show that the stomatal conductances estimated in this work are subject to significant potential inaccuracies in the ERA5 soil water content. These would need to be rectified in future work before a reliable uncertainty analysis can be performed; we will add in the discussion section how such estimates could be quantified.

**Pp 4, lines 126: this motivation of only using the soil moisture of the top layers excluding the information on soil moisture > 1m indicates that you assume that the forests stomatal conductance is mainly controlled by the soil moisture in the top 1m. This might actually depend a lot on the effective rooting depth. I bring this up having seen soil moisture observations in the top soil profiles that seemed to provide a nice source of information to indeed infer the impact of soil water on stomatal conductance but where, then evaluating the observed latent heat fluxes, did not reflect at all observed strong decreases in those soil moisture measurements.**

The decision to only use ERA5 soil water content from depths < 1 m was because the in-situ data used by Li et al (2020) to validate this dataset did not extend beyond this depth (see L126). It is also clear from the FLUXNET comparison exercise that the ERA5 soil moisture dataset may not be adequate for this work, and may need replacing with another soil moisture dataset like H-SAF (2020). However, according to Yang et al (2016), the effective rooting depth of vegetation over much of Europe is < 1 m, so we contend that using soil water content data from lower depths may not be necessary for this work. We will add these points to the final draft.

**Pp 9: lines 175-180; At the end of the methods having seen the overview of all the datasets being used, it makes me wonder about any evaluation strategy that you have developed to at least assess that some of the critical parameters in your inversion of the $O_3$ impact make sense; e.g., did you conduct any evaluation of the Jarvis stomatal conductance based on comparison of the simulated and observed LE? This is a parameter that could have been rather easily evaluated using the FLUXNET datasets.**

We have addressed this problem by comparing the Jarvis stomatal conductance with those estimated from FLUXNET LE measurements in the SynFlux (Ducker et al, 2018), as discussed above.

**And why using the AOT40 where in the previous paragraphs you have referred to the use of Jarvis in the DoseO3 model to evaluate the stomatal dose of O₃? This has to be all better motivated and including the potential implications.**

We use the same model to infer GPP reduction as in Proietti et al (2016) and Anav et al (2011). The choice of using AOT40 was made in order for the units in Equation 6 to cancel out in order to achieve a dimensionless scale factor that can be interpreted as the proportion of GPP lost due to O₃ exposure. The product $g_{sto} \times \text{AOT40}$ in this equation is analogous to the stomatal O₃ dose. We will add this discussion to the final paper draft.

**Figs. 4 and 5: These figures are derived from monthly O3-induced GPP estimates. It would be interesting to show the actual monthly data over the growing season (i.e. a time series), this would provide more insight in the dynamics. The boxplot in Fig. 4 is in fact a bit misleading, as this is typically used to show represent errors/uncertainties, but a proper error propagation is lacking**

We will add a regional time series plot in the final draft of this paper. As we discussed in above, a proper error propagation of the estimated GPP reductions cannot be performed until the issue with the soil water content data is resolved.

| | FLUXNET site | Vegetation type | N | Mean $g_{sto}$ (satellite, cm s$^{-1}$) | Mean $g_{sto}$ (SynFlux, cm s$^{-1}$) | Mean $g_{sto}$ bias (cm s$^{-1}$) | MNMB | FGE | $R^2$ |
|---|---|---|---|---|---|---|---|---|---|
| 1 | CH-Dav | Boreal coniferous | 56 | 0.461 | 0.329 | 0.132 (40.1%) | 0.206 | 0.576 | 0.007 |
| 2 | DE-Lkb | Continental coniferous | 25 | 0.621 | 0.384 | 0.238 (61.9%) | 0.469 | 0.469 | 0.297 |
| 3 | DE-Obe | Continental coniferous | 42 | 0.373 | 0.381 | -0.008 (-2.08%) | -0.072 | 0.290 | 0.073 |
| 4 | DE-SfN | Continental coniferous | 13 | 0.650 | 0.296 | 0.354 (119%) | 0.745 | 0.745 | 0.306 |
| 5 | DE-Tha | Continental coniferous | 36 | 0.373 | 0.256 | 0.117 (45.6%) | 0.311 | 0.373 | 0.372 |
| 6 | DK-Sor | Continental deciduous | 31 | 0.508 | 0.500 | 0.008 (1.54%) | -0.027 | 0.196 | 0.656 |
| 7 | IT-Cp2 | Mediterranean coniferous | 15 | 0.011 | 0.234 | -0.223 (-95.5%) | -1.814 | 1.814 | 0.000 |
| 8 | IT-Cpz | Mediterranean coniferous | 12 | 0.189 | 0.222 | -0.032 (-14.5%) | -0.350 | 0.634 | 0.108 |
| 9 | IT-Lav | Boreal coniferous | 64 | 0.572 | 0.369 | 0.203 (55.0%) | 0.367 | 0.460 | 0.286 |
| 10 | IT-Ren | Boreal coniferous | 58 | 0.162 | 0.402 | -0.240 (-59.7%) | -0.904 | 0.921 | 0.022 |
| 11 | NL-Loo | Continental coniferous | 65 | 0.267 | 0.366 | -0.099 (-27.1%) | -0.359 | 0.396 | 0.394 |

*Table 1: The results of the comparison between FLUXNET SynFlux monthly mean stomatal conductances ($g_{sto}$) and the satellite-based values, for FLUXNET forest sites (see Figure 1). The vegetation type is based on an assessment of the site metadata recorded in the FLUXNET2015 dataset (Pastorello et al, 2020) and the EEA biogeographical region map (EEA, 2016).*

| | FLUXNET site | Temperature | | | VPD | | | Shortwave radiation flux | | | Soil water content | | |
|---|---|---|---|---|---|---|---|---|---|---|---|---|---|
| | | MNMB | FGE | $R^2$ | MNMB | FGE | $R^2$ | MNMB | FGE | $R^2$ | MNMB | FGE | $R^2$ |
| **1** | CH-Dav | -0.483 | 0.514 | 0.888 | -0.293 | 0.295 | 0.825 | 0.035 | 0.069 | 0.772 | 0.431 | 0.431 | 0.027 |
| **2** | DE-Lkb | 0.247 | 0.247 | 0.937 | 0.225 | 0.225 | 0.741 | 0.015 | 0.048 | 0.894 | 0.066 | 0.093 | 0.537 |
| **3** | DE-Obe | 0.121 | 0.128 | 0.950 | 0.112 | 0.176 | 0.774 | 0.042 | 0.081 | 0.965 | 0.240 | 0.253 | 0.096 |
| **4** | DE-SfN | -0.050 | 0.066 | 0.977 | -0.194 | 0.210 | 0.929 | -0.045 | 0.067 | 0.866 | N/A | N/A | N/A |
| **5** | DE-Tha | 0.016 | 0.025 | 0.984 | -0.073 | 0.113 | 0.752 | -0.011 | 0.040 | 0.952 | 0.512 | 0.514 | 0.349 |
| **6** | DK-Sor | 0.056 | 0.059 | 0.977 | -0.019 | 0.057 | 0.849 | 0.039 | 0.051 | 0.862 | 0.194 | 0.200 | 0.163 |
| **7** | IT-Cp2 | -0.057 | 0.057 | 0.978 | -0.305 | 0.305 | 0.766 | 0.040 | 0.043 | 0.947 | -1.584 | 1.584 | 0.103 |
| **8** | IT-Cpz | -0.047 | 0.053 | 0.967 | -0.196 | 0.207 | 0.586 | 0.012 | 0.040 | 0.934 | 0.360 | 0.360 | 0.087 |
| **9** | IT-Lav | 0.185 | 0.185 | 0.972 | 0.247 | 0.247 | 0.798 | -0.146 | 0.147 | 0.914 | -0.229 | 0.246 | 0.263 |
| **10** | IT-Ren | 0.263 | 0.263 | 0.966 | 0.418 | 0.418 | 0.689 | -0.059 | 0.078 | 0.767 | -0.642 | 0.660 | 0.015 |
| **11** | NL-Loo | 0.013 | 0.016 | 0.995 | -0.018 | 0.070 | 0.883 | -0.024 | 0.052 | 0.854 | 0.990 | 0.990 | 0.260 |

*Table 2: The results of the comparison between the FLUXNET2015 monthly means of variables related to stomatal conductance, and the values taken from ERA5, for FLUXNET forest sites (see Figure 1). No soil water content data was available from DE-SfN.*

[Figure]

*Figure 1: The results of the comparison between FLUXNET SynFlux monthly mean stomatal conductances ($g_{sto}$) and the satellite-based values, for FLUXNET forest sites (see Table 1). The locations of the DE-Obe and IT-Cpz stations were shifted to improve readability. (top left) Mean monthly $g_{sto}$ reported by SynFlux between April – September, 2003 – 2015. Also shown are the MNMB (top right), FGE (bottom left), and $R^2$ (bottom right) inferred from the satellite-SynFlux comparisons.*

References

Anav, A., Proietti, C., Menut, L., Carnicelli, S., De Marco, A., and Paoletti, E.: Sensitivity of stomatal conductance to soil moisture: implications for tropospheric ozone, Atmos. Chem. Phys., 18, 5747–5763, https://doi.org/10.5194/acp-18-5747-2018, 2018.

Anav, A., Menut, L., Khvorostyanov, D., and Viovy, N.: Impact of tropospheric ozone on the Euro-Mediterranean vegetation, Global Change Biology, 17, 2342–2359, https://doi.org/10.1111/j.1365-2486.2010.02387.x, 2011.

Bennouna, Y., Schulz, M., Christophe, Y., Eskes, H., Basart, S., Benedictow, A., Blechschmidt, A.-M., Chabrillat, S., Clark, H., Cuevas, E., and et al.: Validation report of the CAMS global Reanalysis of aerosols and reactive gases, years 2003-2019, Copernicus Atmosphere Monitoring Service, https://doi.org/10.24380/2V3P-AB79, https://atmosphere.copernicus.eu/sites/default/files/2020-04/CAMS84_2018SC2_ 360 D5.1.1-2019.pdf, 2020.

EEA: Biogeographical regions in Europe, European Environment Agency, available from: https://www.eea.europa.eu/data-andmaps/data/biogeographical-regions-europe-3, 2016.

Ducker, J. A., Holmes, C. D., Keenan, T. F., Fares, S., Goldstein, A. H., Mammarella, I., Munger, J. W., and Schnell, J.: Synthetic ozone deposition and stomatal uptake at flux tower sites, Biogeosciences, 15, 5395–5413, https://doi.org/10.5194/bg-15-5395-2018, 2018.

H-SAF, Scatterometer root zone soil moisture (RZSM) data record 10 km resolution – multimission. Darmstadt, Germany: Eumetsat. Available at: https://navigator.eumetsat.int/product/EO:EUM:DAT:0231, 2020.

Hersbach, H., Bell, B., Berrisford, P., Hirahara, S., Horányi, A., Muñoz-Sabater, J., Nicolas, J., Peubey, C., Radu, R., Schepers, D., Simmons, A., Soci, C., Abdalla, S., Abellan, X., Balsamo, G., Bechtold, P., Biavati, G., Bidlot, J., Bonavita, M., De Chiara, G., Dahlgren, P., Dee, D., Diamantakis, M., Dragani, R., Flemming, J., Forbes, R., Fuentes, M., Geer, A., Haimberger, L., Healy, S., Hogan, R. J., Hólm, E., Janisková, M., Keeley, S., Laloyaux, P., Lopez, P., Lupu, C., Radnoti, G., de Rosnay, P., Rozum, I., Vamborg, F., Villaume, S., and Thépaut, J.-N.: The ERA5 Global Reanalysis, Q. J. Roy. Meteor. Soc., 146, 730, 1999–2049, https://doi.org/10.1002/qj.3803, 2020.

Huijnen, V., Miyazaki, K., Flemming, J., Inness, A., Sekiya, T., and Schultz, M. G.: An intercomparison of tropospheric ozone reanalysis products from CAMS, CAMS interim, TCR-1, and TCR-2, Geosci. Model Dev., 13, 1513–1544, https://doi.org/10.5194/gmd-13-1513-2020, 2020.

Inness, A., Ades, M., Agustí-Panareda, A., Barré, J., Benedictow, A., Blechschmidt, A.-M., Dominguez, J. J., Engelen, R., Eskes, H., Flemming, J., Huijnen, V., Jones, L., Kipling, Z., Massart, S., Parrington, M., Peuch, V.-H., Razinger, M., Remy, S., Schulz, M., and Suttie, M.: The CAMS reanalysis of atmospheric composition, Atmos. Chem. Phys., 19, 3515–3556, https://doi.org/10.5194/acp-19-3515-2019, 2019.

Li, M., Wu, P., and Ma, Z.: A comprehensive evaluation of soil moisture and soil temperature from third-generation atmospheric and land reanalysis data sets, International Journal of Climatology, https://doi.org/10.1002/joc.6549, 2020.

Lombardozzi, D., Levis, S., Bonan, G., Hess, P. G., and Sparks, J. P.: The Influence of Chronic Ozone Exposure on Global Carbon and Water Cycles, J. Climate, 28, 292–305, doi:10.1175/JCLI-D14-00223.1, 2015.

Mao, J. and Yan, B.: Global Monthly Mean Leaf Area Index Climatology, 1981-2015. ORNL DAAC, https://doi.org/10.3334/ORNLDAAC/1653.

Pastorello, G., Trotta, C., Canfora, E., et al.: The FLUXNET2015 dataset and the ONEFlux processing pipeline for eddy covariance data, Scientific Data, 7, 225, https://doi.org/10.1038/s41597-020-0534-3, 2020.

Proietti, C., Fornasier, M. F., Sicard, P., Anav, A., Paoletti, E., and Marco, A. D.: Trends in tropospheric ozone concentrations and forest impact metrics in Europe over the time period 2000–2014, Journal of Forestry Research, 32, 543–551, https://doi.org/10.1007/s11676- 020-01226-3, 2020.

Running, S. W. and Zhao, M.: User's Guide: Daily GPP and annual NPP (MOD17 A2/A3) products, NASA Earth Observing System MODIS land algorithm 1–28, 2015.

Wagner, A., Bennouna, Y., Blechschmidt, A. M., Brasseur, G., Chabrillat, S., Christophe, Y., et al.: Comprehensive evaluation of the Copernicus Atmosphere Monitoring Service (CAMS) reanalysis against independent observations: Reactive gases, Elementa: Science of the Anthropocene, 9, 1, 00171, https://doi.org/10.1525/elementa.2020.00171, 2021.

Yang, Y., Donohue, R. J., and McVicar, T. R.: Global estimation of effective plant rooting depth: Implications for hydrological modeling, Water Resour. Res., 52, 8260–8276, https://doi.org/10.1002/2016WR019392, 2016.

---

## Author Comment (AC2)

**Response to RC2 comments**

We thank the reviewer for their comments, and will respond to them herein.

**Section 2.2.1 & 2.2.2: did you use (extrapolate) meteorological and ozone data at canopy height (ca. 20-25 m)? as you used the parameterization of the DO3SE model for sunlit leaves at the top canopy (Table 1).**

We did not perform any extrapolation of the data to the canopy height, as insufficient data at the time of this work (e.g. canopy surface area) was available to reliably do this. As well as this, the CAMS $O_3$ lowermost profile layer covers the region above the boundary layer as well, so the $O_3$ concentrations can be considered as being representative of the canopy height. Proietti et al (2016) similarly did not scale their in-situ measured $O_3$ concentrations to canopy height, and still obtained results comparable to prior studies. Future iterations of this work will attempt to use satellite-derived canopy height datasets such as those derived from the GEDI instrument (Potapov et al, 2021). We will add this explanation to the final paper draft.

**Table 1: mistake for Tmin for Mediterranean species (Deciduous Tmin = 0; Evergreen Tmin = 1). Explain why Tmax is set at 200°C.**

We thank the reviewer for pointing this error out, and will adjust Table 1 in the final draft. According to Mills et al (2017), $T_{max}$ = 200 °C for boreal vegetation to simulate the weak response to high temperatures of Norway spruce and birch trees growing under Northern European conditions (here, the stomatal response is instead restricted by high VPD values). Therefore, this $T_{max}$ value should be treated as a forcing instead of a descriptive parameter. This explanation will also be added to the final paper draft.

**L114: why an overestimation of 15% is observed in Southern Europe?**

Since the original submission of this paper, Wagner et al (2021) published a more detailed comparison of CAMS and in-situ $O_3$. They found that while CAMS reproduces tropospheric $O_3$ within 10% of independent observations, a seasonal mean variability in biases exist over Northern midlatitudes, peaking at 15% in October.

Over Northern Europe, and to a lesser extent Southern Europe, it was found that the CAMS overestimation of surface $O_3$ was greater during nighttime than daytime hours. They propose that this indicates that nocturnal $O_3$ destruction processes in the boundary layer (e.g. $NO_x$ titration) are not being correctly represented in the global model, which is known to have difficulties resolving subgrid processes.

We will update the final draft of this paper with the above explanation.

**L167: what is the layer depth for SWC? Soil moisture is usually obtained for the upper 10-20 cm of soil, which resulted in a worst-case risk scenario, as the uppermost soil layers are expected to dry out more easily than deeper layers.**

The ERA5 soil layer depths are as follows: Layer 1: 0-7cm, Layer 2: 7-28cm, Layer 3: 28-100cm, Layer 4: 100-289cm (Hersbach et al, 2020). By using only the first three layers, we take the mean soil water content between 0-1 m. We will add this detail to the final paper draft.

**L195: which test did you use for trend analysis: Mann-Kendall test, Sen method?**

We use the Wald test with a t-test distribution, assuming a null hypothesis that the gradient is zero. This information will be added to the final draft.

**Figure 8: why some areas are missing (e.g., the UK, Southeastern Spain, Northwestern France)?**

As stated in L140, the ESA-CCI dataset used to determine the vegetation type of each grid cell needed to be resampled to the ERA5 0.25° × 0.25° spatial grid using majority resampling (i.e. grid cells were assigned the most common ESA-CCI classification value). The forests in the regions mentioned by the reviewer are very fragmented, with many surrounded by much larger cropland or grassland. As a result, many grid cells in these areas were classified as not being forested, and were therefore were not analysed in this work.

In the future, we will ensure that forests in these regions are better represented. One way to do this would be to replace the ERA5 meteorological data with ERA5-Land (Muñoz-Sabater et al, 2021), which has a higher spatial resolution.

**L235: do you mean "tropospheric" rather than "anthropogenic"? Surface ozone can be formed from biogenic VOCs.**

We agree that "tropospheric" would be a better word to use here, and will correct this line as suggested in the final draft.

References

Hersbach, H., Bell, B., Berrisford, P., Hirahara, S., Horányi, A., Muñoz-Sabater, J., Nicolas, J., Peubey, C., Radu, R., Schepers, D., Simmons, A., Soci, C., Abdalla, S., Abellan, X., Balsamo, G., Bechtold, P., Biavati, G., Bidlot, J., Bonavita, M., De Chiara, G., Dahlgren, P., Dee, D., Diamantakis, M., Dragani, R., Flemming, J., Forbes, R., Fuentes, M., Geer, A., Haimberger, L., Healy, S., Hogan, R. J., Hólm, E., Janisková, M., Keeley, S., Laloyaux, P., Lopez, P., Lupu, C., Radnoti, G., de Rosnay, P., Rozum, I., Vamborg, F., Villaume, S., and Thépaut, J.-N.: The ERA5 Global Reanalysis, Q. J. Roy. Meteor. Soc., 146, 730, 1999–2049, https://doi.org/10.1002/qj.3803, 2020.

Muñoz-Sabater, J., Dutra, E., Agustí-Panareda, A., Albergel, C., Arduini, G., Balsamo, G., Boussetta, S., Choulga, M., Harrigan, S., Hersbach, H., Martens, B., Miralles, D. G., Piles, M., Rodríguez-Fernández, N. J., Zsoter, E., Buontempo, C., and Thépaut, J.-N.: ERA5-Land: a state-of-the-art global reanalysis dataset for land applications, Earth Syst. Sci. Data, 13, 4349–4383, https://doi.org/10.5194/essd-13-4349-2021, 2021.

Potapov, P., Li, X., Hernandez-Serna, A., Tyukavina, A., Hansen, M. C., Kommareddy, A., et al.: Mapping global forest canopy height through integration of GEDI and Landsat data, Remote Sensing of Environment, 253, 112165. https://doi.org/10.1016/j.rse.2020.112165, 2021.

Proietti, C., Anav, A., De Marco, A., Sicard, P., and Vitale, M.: A multi-sites analysis on the ozone effects on Gross Primary Production of European forests, Science of The Total Environment, 556, 1 – 11, https://doi.org/10.1016/j.scitotenv.2016.02.187, 2016.

Wagner, A., Bennouna, Y., Blechschmidt, A. M., Brasseur, G., Chabrillat, S., Christophe, Y., et al.: Comprehensive evaluation of the Copernicus Atmosphere Monitoring Service (CAMS) reanalysis against independent observations: Reactive gases, Elementa: Science of the Anthropocene, 9, 1, 00171, https://doi.org/10.1525/elementa.2020.00171, 2021.